# Genomics of Plasma Cell Leukemia

**DOI:** 10.3390/cancers14061594

**Published:** 2022-03-21

**Authors:** Elizabeta A. Rojas, Norma C. Gutiérrez

**Affiliations:** 1Hematology Department, University Hospital of Salamanca, Institute of Biomedical Research of Salamanca (IBSAL), 37007 Salamanca, Spain; elirr@usal.es; 2Cancer Research Center-Institute of Cancer Molecular and Cellular Biology (CIC-IBMCC) (USAL-CSIC), 37007 Salamanca, Spain; 3Centro de Investigación Biomédica en Red de Cáncer (CIBERONC), CB16/12/00233, 28029 Madrid, Spain; 4Grupo Español de Mieloma (GEM), 28040 Madrid, Spain

**Keywords:** plasma cell leukemia, PCL, genetics, primary and secondary PCL, multiple myeloma, mutations, transcriptome

## Abstract

**Simple Summary:**

Plasma cell leukemia (PCL) is a very aggressive plasma cell disorder with a dismal prognosis, despite the therapeutic progress made in the last few years. The implementation of genomic high-throughput technologies in the clinical setting has revealed new insights into the genomic landscape of PCL, some of which may have an impact on the development of novel therapeutic approaches. The purpose of this review is to provide a comprehensive overview and update of the genomic studies carried out in PCL.

**Abstract:**

Plasma cell leukemia (PCL) is a rare and highly aggressive plasma cell dyscrasia characterized by the presence of clonal circulating plasma cells in peripheral blood. PCL accounts for approximately 2–4% of all multiple myeloma (MM) cases. PCL can be classified in primary PCL (pPCL) when it appears de novo and in secondary PCL (sPCL) when it arises from a pre-existing relapsed/refractory MM. Despite the improvement in treatment modalities, the prognosis remains very poor. There is growing evidence that pPCL is a different clinicopathological entity as compared to MM, although the mechanisms underlying its pathogenesis are not fully elucidated. The development of new high-throughput technologies, such as microarrays and new generation sequencing (NGS), has contributed to a better understanding of the peculiar biological and clinical features of this disease. Relevant information is now available on cytogenetic alterations, genetic variants, transcriptome, methylation patterns, and non-coding RNA profiles. Additionally, attempts have been made to integrate genomic alterations with gene expression data. However, given the low frequency of PCL, most of the genetic information comes from retrospective studies with a small number of patients, sometimes leading to inconsistent results.

## 1. Introduction

Plasma cell leukemia (PCL) is an uncommon plasma cell dyscrasia with an aggressive course and poor prognosis. PCL represents less than 3% of all plasma cells neoplasms, and its incidence has been estimated at 0.04 cases per 100,000 persons/year [1,2].

Historically, PCL has been defined by the presence of more than 20% of circulating plasma cells (PCs) and an absolute number of ≥2 × 10^9^/L of PCs in peripheral blood [3]. However, in some studies, only the presence of one of these criteria had been considered to define PCL. Moreover, recent studies have shown that much lower levels of circulating PC have the same adverse prognostic impact. Accordingly, the consensus recently published by the International Myeloma Working Group (IMWG) states that PCL is defined by the presence of 5% or more circulating plasma cells in peripheral blood [4].

PCL is classified as primary PCL (pPCL) when it occurs de novo so that the patient has no evidence of previous multiple myeloma (MM), and as secondary PCL (sPCL) when leukemic progression occurs in the context of pre-existing refractory or relapsing MM [5,6]. pPCL is more frequent than sPCL, representing about 60–70% of patients, [7] and occurs in patients significantly younger than sPCL. Nevertheless, the number of sPCL cases has been increasing in recent years, which is probably related to the increased survival of MM patients.

The clinical presentation of PCL is more aggressive than that observed in MM, including more severe cytopenias, hypercalcemia, and renal insufficiency. Higher tumor burden and proliferation activity of PCL are manifested by greater levels of B2-microglobulin and lactate dehydrogenase (LDH). Extramedullary involvement (lymph nodes, liver, spleen, pleura, and central nervous system) at diagnosis is more common in pPCL and sPCL than in MM, but osteolytic lesions are more frequent in sPCL and MM than in pPCL [8,9,10,11,12,13].

Various studies have analyzed the immunophenotype of PCL. The two common PCs markers, CD38 and CD138 antigens, are similarly expressed in MM and PCL. However, PCL displays a more immature phenotype than MM, expressing more frequently CD20, CD23, CD28, CD44, and CD45, and less frequently CD9, CD56, CD71, CD117, and HLA-DR antigens [14,15,16].

PCL patients are characterized by short remissions and early relapses. The 5-year survival rate from the diagnosis of PCL does not exceed 10%. Survival of sPCL patients is consistently shorter than pPCL [8]. The incorporation of new therapeutic agents has not achieved significant improvements in the survival of PCL, unlike what has been attained in MM. The low incidence of PCL makes it difficult to conduct studies aimed at exploring the efficacy of new drugs that would eventually help to establish an optimal therapeutic option. Thus, therapeutic recommendations are supported by small prospective and retrospective studies and sometimes by data extrapolated from clinical trials with MM patients. The therapeutic strategy usually followed in transplant-candidate patients generally includes an intensive induction with bortezomib-based regimens also containing lenalidomide and chemotherapeutic agents. After autologous stem cell transplantation (ASCT), there is increasing consensus on continuing a consolidation and maintenance therapy, although the therapeutic agents that should be included are not well established. A tandem transplant with an ASCT followed by reduced-intensity allogenic transplantation can also be considered. However, even using the most intensive therapeutic arsenal, the prognosis of PCL remains ominous. There is, therefore, a compelling need to advance in the search for new drugs with different mechanisms of action and more closely related to the genetic features of tumor PC. In this regard, the development of *BCL2* inhibitors and the new immunotherapeutic approaches, such as chimeric antigen receptor T-cells (CAR-T cells) and monoclonal and bispecific antibodies, opens up new opportunities in the treatment of PCL patients.

Early cytogenetic studies performed in PCL had already revealed some differences between this entity and MM. Later on, the widespread use of fluorescence in situ hybridization (FISH) has increased our knowledge of genetic alterations. In recent years, the development of high-throughput genomic analysis tools has helped to better understand the genetic particularities of PCL. However, the robustness of the results is undermined by the limited number of patients included in the studies because of the low incidence of this disease. In this review, we mainly focus on the genomic characteristics of pPCL, although some data concerning sPCL are provided when considered of interest. A summary of the most relevant results provided by the main genomic studies carried out in PCL is shown in Table 1.

## 2. Cytogenetic Abnormalities

Early cytogenetic and DNA content studies carried out in PCL revealed that there was a predominance of non-hyperdiploid cases (more than 50% of pPCL) compared to that observed in MM [8,14,18]. These results were confirmed in subsequent studies using not only conventional karyotyping but also molecular cytogenetic techniques such as comparative genomic hybridization (CGH) and single nucleotide polymorphism (SNP)-arrays, which showed that pPCL had more DNA copy number changes with a predominance of chromosomal losses in contrast to MM [19,27]. As in MM, FISH has been routinely carried out to identify cytogenetic alterations present in pPCL at the time of diagnosis. Virtually all the studies reporting data provided by FISH analysis, sometimes in combination with other cytogenetic techniques, point out that the chromosomal abnormalities observed in pPCL are mostly the same recurrently found in MM, although many of them are present with greater frequency (Figure 1).

Monosomy and deletions of chromosome 13 (del(13q)) have been observed in approximately 85% of pPCL [8,22,35]. Abnormalities of chromosome 1 are also frequent in pPCL patients. Gain (3 copies) and amplification (≥4 copies) of chromosome arm 1q21 (gain/amp(1q)) have been reported in around 70% of pPCL cases [35,36]. Although the frequency of gain/amp(1q) does not reach such a high percentage in newly diagnosed MM patients, the incidence of this abnormality increases in relapsed/refractory MM up to 50–80% [22,43,44]. Likewise, most of the studies have shown greater frequency of deletion of 1p (del(1p)) in pPCL than in MM patients (24–33% vs. 9–18%, respectively) [22,41]. While the impact of abnormalities in chromosome 1, both gain/amp(1q) and del(1p), on the survival of patients with MM is well established [45,46], their effect on the prognosis of pPCL is still poorly substantiated. Only one study has reported that del(1p), but not gain/amp(1q), is associated with shorter survival of PCL patients, although the set of sPCL included in the study may be biasing the influence that this chromosomal alteration might have on pPCL considered as a separate entity [22]. Deletion of 17p (del(17p)), although uncommon in MM at the time of diagnosis, reaches frequencies of 50% in pPCL [8,22,27,32,37]. However, it seems to have no impact on the prognosis of pPCL, unlike in MM [8,22].

Taken together, all these results showing the increasing frequency of the aforementioned chromosome imbalances from MM to PCL support the multistep transformation model from monoclonal gammopathy of undetermined significance (MGUS) through smoldering multiple myeloma (SMM) and MM to PCL that leads to progressive accumulation of secondary genetic alterations.

The incidence of *IGH* translocations is significantly higher in pPCL than in MM. Several studies show that t(11;14) leading to *CCND1* dysregulation are significantly more frequent in pPCL than in MM, reaching percentages as high as 45–70% in some series [8,17,18,21,22,23,24,27,40,41]; also noteworthy is the high proportion of t(14;16) detected in pPCL compared with MM (13–25% vs. 1–5%, respectively), which is supported by five studies [18,21,23,36,41]. Conversely, in most of the studies, t(4;14) has been found to be less frequent in pPCL than in MM [23,36,38].

The t(11;14) has largely been demonstrated to be a neutral prognostic factor for MM survival [47]. Although no influence of t(11;14) in the survival of pPCL patients was initially observed [8,22], it has recently been reported that pPCL patients bearing t(11;14) had a significantly longer OS than those without this abnormality [42]. On the contrary, t(4;14) has been associated with poor prognosis [22].

*MYC* rearrangements have also been found in PCL, although the reported incidence varies between 13% and 40% [8,40,48]. Moreover, an association between *MYC* rearrangement and shorter overall survival of pPCL patients has been shown [8].

Other chromosomal abnormalities have been identified in pPCL, especially the loss of chromosome 16 (80%) [19,23,49], 7 (11%) [50] and X (25%) [14], and the trisomy of chromosome 8 (43%) [14].

## 3. Gene Mutations

Before the availability of next-generation sequencing (NGS) technologies, the mutational status of *RAS* oncogenes (*NRAS* and *KRAS*), the two most prevalent mutated genes in MM, and of the tumor suppressor *TP53*, had been explored in pPCL using traditional DNA sequencing methodologies. Two studies demonstrated a high incidence of *NRAS* and *KRAS* activating mutations: one of them reported these mutations at codons 12, 13, or 61 in 27% of pPCL and 15% of sPCL cases [8], and in the other study *NRAS* and/or *KRAS* mutations were found in 50% of pPCL cases and in 55% of MM [20]. Strikingly, these findings were not confirmed in a subsequent study [27]. *TP53* is one of the most frequently mutated genes in pPCL in all the published series, reaching frequencies of 25% [8,23,27]. The proportion of cases with biallelic inactivation of *TP53* is also greater in pPCL than in MM (17–35% vs. 3–4%) [8,27]. *TP53* coding mutations involving 5–8 exons were found, predicting all of them a non-functional p53 protein [8,27] (Figure 1).

The first whole-exome sequencing (WES) analysis of pPCL revealed a highly heterogeneous mutational profile [29]. Almost 2000 coding somatic non-silent variants on 1643 genes were described, with more than 160 variants per sample, although with hardly any recurrent mutations in two or more samples. Fourteen mutated genes mainly involved in cell cycle and apoptosis (*CIDEC*), RNA binding and degradation (*DIS3*, *RPL17*), and cell-matrix adhesion and membrane organization (*SPTB*, *CELA1*) were considered as potential cancer driver genes in pPCL. Other studies have confirmed that the number of nonsynonymous mutations per sample is higher in pPCL than in MM [36].

As in MM, activating *N*/*KRAS* mutations have been identified in pPCL using WES methodologies, although the proportions were significantly unequal between the two of the studies. The first study reported mutations of *KRAS* and *NRAS* only in two distinct samples (<10% of the pPCL). This study highlighted that *KRAS* and *NRAS* were three-fold less frequently mutated in pPCL compared to that observed in MM [29]. On the contrary, the second study also using WES methodology found that *KRAS* was the most frequently mutated gene in pPCL samples (around 39%), and mutations of *NRAS* were present in 13% of pPCL [36]. Using targeted NGS approaches, *KRAS* mutations were detected in 17% of pPCL, 18% of sPCL, and 33% of MM, and *NRAS* mutations in 4% of pPCL, 36% of sPCL, and 27% of MM [30]. Apparently, the *MEK*/*ERK* signaling pathway was less affected by mutation events in pPCL than in sPCL and MM [30].

Mutations of the *BRAF* gene have also been detected in pPCL samples. A low frequency and even absence of *BRAF* mutations in pPCL patients have been described using WES [29,36]. However, when targeted NGS was applied, the frequency of *BRAF* mutations detected in pPCL was higher (21% in pPCL and 9% in sPCL). It is worth highlighting the role that the different coverage levels among NGS studies and the small number of patients analyzed may be playing in the conflicting results.

*TP53* gene has also been analyzed by NGS in pPCL [32,36,41], confirming the results previously observed using traditional DNA sequencing methodologies, namely, the high proportion of *TP53* mutations in pPCL. Interestingly, the presence of *TP53* mutations has been associated with significantly shorter survival in the study, including the largest number of patients with pPCL to date [41]. *IRF4* mutations have recently been shown to be significantly more frequent in pPCL than in MM patients (11% vs. 4%) [41]. Other gene mutations commonly observed in MM have also been reported in pPCL but with different frequencies. Schinke et al. detected *DIS3* and *PRMD1* mutations in 5% and 13% of patients with pPCL, respectively, while Cifola et al. identified *DIS3* mutations in 25% of cases and no variants in the *PRMD1* gene. Both studies have described a similar incidence of *FAM46C* mutation (10–12%) in pPCL patients [29,36].

## 4. Transcriptome Characterization

Several studies have explored the gene expression profile (GEP) analyzed by microarrays in pPCL. All of them have identified a transcriptome signature characteristic of pPCL and different from that of MM. The first two reports identified a gene-specific signature that distinguished pPCL from MM cases, although the number of overlapped genes between both datasets containing the differentially expressed genes was only around 15%. The functional annotation analysis identified dysregulation of lipid metabolism, glucocorticoid receptor, and *IL6* pathways in one study [25], and alterations of *NF-kB* pathway, *FAS* signaling, structural organization of the cell and migration processes in the other study [28]. A transcriptional signature including 27 genes has been associated with the overall survival of pPCL, despite the cytogenetic alterations. Interestingly, none of these genes had been selected in MM-high risk signatures [28].

More recently, the GEP of 41 pPCL patients has been compared to that of more than 700 newly diagnosed MM [36]. In pPCL, the analysis showed overexpression of genes previously related to MM biology or prognosis, such as *PHF19* and *TAGLN2*, and underexpression of the adhesion molecules *VCAM1* and *CD163*, which are highly expressed in MM and have been correlated with poor survival [51,52].

RNA-seq analysis of pPCL has also shown a specific transcriptional landscape of pPCL, as previously demonstrated by GEP using microarrays. Compared to MM, pPCL showed significantly higher expression of genes involved in G2M checkpoint and *MYC* target genes and lower expression of genes involved in p53 pathway, hypoxia, and *TNF alpha* signaling via *NF-κB* [41]. In this regard, significant overexpression of *CDKN2A, CCND3*, and *CCND1* genes, using quantitative RT-PCR, has been reported in PCL compared to MM samples, indicating a marked cell cycle dysregulation in the transition from MM to PCL [53].

A comprehensive molecular analysis of pPCL integrating data from FISH, SNP-arrays, and GEP has revealed a strong correlation between chromosomal imbalances and transcriptional modulation. The gene dosage effect was particularly observed in those genes mapping 1q chromosome [27]. In addition, the analysis of upregulated and downregulated transcripts in the gained and lost chromosomal regions, respectively, found that protein transport, translation, and biosynthesis functional categories were upregulated in pPCL cases with gained chromosomal regions, whereas RNA splicing, protein catabolic process, and regulation of apoptosis were downregulated in pPCL cases with deleted regions.

Differences between the gene expression signature of pPCL and MM could be partly attributed to the dissimilar distribution of genetic abnormalities between the two diseases. This fact prompted us to compare the transcriptome of pPCL and MM patients using samples with del(17p) and a similar cytogenetic background [34]. This approach revealed that pPCL and MM were separated into two differentiated clusters despite the equivalent cytogenetic profile shared by both entities. Differentially expressed genes were mostly downregulated in pPCL, among which were genes associated with bone marrow microenvironment and bone diseases in MM, such as *DKK1*, *KIT*, *NCAM1*, and *FRZB* (Figure 1). Interestingly, the analysis focused on isoform expression showed that dysregulation of RNA splicing machinery may be a relevant molecular mechanism underlying the biological differences between the pPCL and MM.

A similar approach has been used to ascertain the differences in the transcriptome between pPCL and MM samples harboring t(11;14) [38]. In line with our results, this study shows that both plasma cell dyscrasias are clearly distinguishable based on the transcriptome profile despite sharing a uniform genetic background. pPCL with t(11;14) were positively associated with genes involved in *IL2-STAT5* signaling but negatively associated with the regulation of cell and cell adhesion pathways. In any case, the most relevant finding of this study was that pPCL showed a different expression pattern of the *BCL2* family genes and of the B-cell-associated genes, despite the presence of t(11;14) in both PCL and MM samples. These results suggest that the efficacy of venetoclax in pPCL and MM patients with t(11;14) may be associated with different molecular programs.

## 5. Non-Coding RNA Profile

Non-coding RNAs (ncRNAs) are classified as short (<200 nucleotides) and long (>200 nucleotides). The miRNAs are short ncRNAs of 19–22 nucleotides that regulate gene expression at the post-transcriptional level. Since their discovery, numerous studies have attributed a wide variety of functions for ncRNAs in the pathogenic mechanisms of MM [54,55,56].

There is only one study analyzing the expression pattern of miRNAs in pPCL [26]. The analysis of 18 pPCL identified 42 upregulated and 41 downregulated miRNAs in pPCL when compared with MM samples. Moreover, seven miRNAs were found to be differentially expressed depending on the type of *IGH* translocation. Three miRNAs (let-7e, miR-135a, and miR-148a) were overexpressed in PCL patients with t(4;14); three (miR-7, miR-7-1, and miR-454) underexpressed in PCL with t(14;16); and the miR-342-3p was underexpressed in PCL with t(11;14). Notably, four miRNAs, miR-22, miR-146a, miR-92a, and miR-330-3p, were found to have an impact on the survival of pPCL patients. The overexpression of miR-146a, which was associated with shorter progression-free survival (PFS) in pPCL cases, and miR-22, which was associated with longer PFS, showed a pro- and anti-survival effect, respectively, in myeloma cell lines [26]. Accordingly, one study has demonstrated that MM cells stimulate the overexpression of miR-146a in mesenchymal stromal cells, resulting in more cytokine secretion and enhancing cell viability of MM cells [57] (Figure 1).

Long non-coding RNAs (lncRNAs) are a group of very heterogeneous non-coding RNAs with a length of more than 200 nucleotides. They have a similar structure to mRNAs but are not translated to functional proteins. LncRNAs represent more than 50% of the non-coding RNAs, and their functions are related to the regulation of transcription, genome integrity, cell differentiation, X-chromosome inactivation, and development, among others [58].

LncRNAs expression profile has also been investigated in a large cohort of PC dyscrasias, including samples from MGUS, SMM, MM, and PCL together with NPC [31]. Differential expression of 160 lncRNAs between NPC and the four premalignant and malignant entities was detected. In particular, expression levels of 15 lncRNAs were progressively increased from NPC to PCL patients, while six lncRNAs showed a significant decrease in the transition from NPC and premalignant entities to more aggressive forms. LncRNAs involved in the progression from MM to PCL have recently been explored [39]. A total of 13 dysregulated lncRNAs was detected. A significant underexpression of lymphocyte antigen antisense RNA 1 (*LY86-AS1*) and VIM antisense RNA 1 (*VIM-AS1*) was observed in PCL compared to MM and further validated by qRT-PCR. However, their functions in MM to PCL progression remain unknown.

Differential expression of lncRNAs has also been detected between pPCL and MM samples with t(11;14) [38]. In particular, the lncRNA *SNHG6*, whose overexpression was associated with significantly inferior overall survival in MM patients from the CoMMpass dataset, was found to be upregulated in the pPCL patients.

## 6. Methylation Patterns

The analysis of global methylation patterns in pPCL using high-density arrays has identified a global hypomethylation profile in pPCL samples [33] (Figure 1). The comparison of methylation levels between pPCL, MM, MGUS, and NPC samples revealed that genes highly methylated in NPC underwent a progressive decrease in the levels of methylation as the aggressiveness of the disease increased from MGUS to MM and pPCL. Curiously, pPCL patients showed distinct methylation profiles depending on the presence of *DIS3* gene mutations, t(11;14), and t(14;16). On the contrary, Walker et al. [59] had previously found gene-specific hypermethylation of almost 2000 genes in the transition from MM to PCL, although the number of PCL cases was quite small.

## 7. Concluding Remarks

Chromosomal, genetic, and genomic alterations found in pPCL are sufficiently different from those observed in MM to consider it a distinct clinicopathological entity and not merely a more aggressive form of MM. However, the low incidence of this disease makes it extremely difficult to gather enough pPCL cases to carry out genomic studies that provide consistent results. On the other hand, the paucity of clinical trials specifically designed for this disease precludes prospective studies. In this regard, proposals aimed at collecting hundreds of pPCL samples involving numerous centers in order to conduct biological studies could represent a breakthrough in identifying dysregulations of signaling pathways that could be therapeutically targeted.

## Figures and Tables

**Figure 1 cancers-14-01594-f001:**
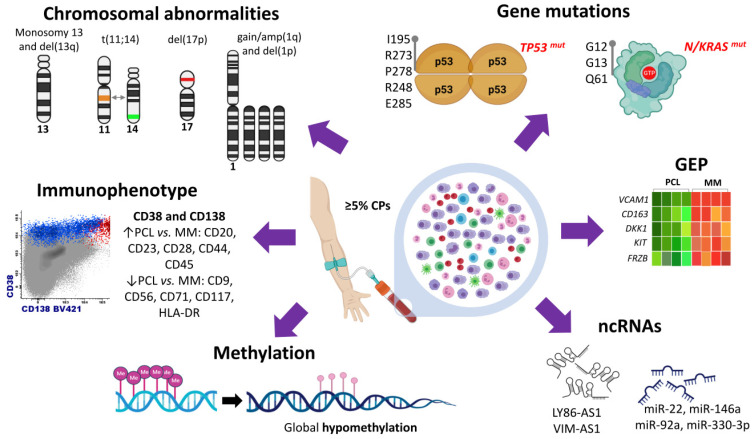
Genomic abnormalities of primary plasma cell leukemia (pPCL). The updated consensus of the IMWG defines pPCL by the presence of 5% or more circulating plasma cells in peripheral blood. Cytogenetic studies by FISH show predominance of monosomy and deletions of chromosome 13, t(11;14), del(17p), gain/amp(1q) and del(1p). Mutation studies by conventional DNA sequencing, WES, and targeted NGS detect a high frequency of mutations in *TP53* and *K/NRAS* genes. The amino acids most frequently mutated in *TP53* are I195, R273, P278, R248, and E285. Activating mutations of *K/NRAS* most frequently found in pPCL patients affect codons 12, 13, and 61 (G12, G13, and Q61). Immunophenotyping of plasma cells reveals expression of CD38 and CD138 in both pPCL and MM, although higher expression of CD20, CD23, CD28, CD44, and CD45 and lower expression of CD9, CD56, CD71, CD117, and HLA-DR may be found in pPCL compared to MM. Gene expression profiling in pPCL has shown downregulation of genes associated with bone marrow microenvironment and bone diseases in MM, such as *DKK1*, *KIT*, and *NCAM1* genes. A global hypomethylation profile has been found in pPCL samples. Non-coding RNAs (miRNAs and lncRNAs) are dysregulated in pPCL, and some of them are associated with survival of patients (as shown in the figure).

**Table 1 cancers-14-01594-t001:** Summary of the most relevant genomic studies carried out in PCL.

Study/Reference	Number of Patients	Methodologies	Summary of Results *
**Avet-Loiseau et al., 1998** [17]	14 pPCL/127 MM	FISH	*IGH* translocations in 71% pPCL.
**García-Sanz et al., 1999** [14]	26 pPCL/664 MM	Cell DNA content, immunophenotypic studies, FISH	Numeric abnormalities in 92% pPCL. DNA content: diploid in 85% pPCL.
**Avet-Loiseau et al., 2001** [18]	40 pPCL/247 MM	FISH, conventional karyotyping	Higher proportion of t(11;14), t(14;16), and hypodiploid karyotype in pPCL.
**Gutiérrez et al.,** **2001** [19]	5 pPCL/25 MM	CGH	Losses of chromosomal material significantly more frequent in pPCL.
**Bezieau et al.,** **2001** [20]	10 pPCL/3 sPCL/33 MM/6 MGUS/2 SMM/11 MM at relapse	Allele-specific PCR amplification and *K/NRAS* direct sequencing	*K/NRAS* mutations in 55% MM at diagnosis, 81% MM at relapse, and 50% pPCL. *KRAS* mutations were always more frequent than *NRAS*.
**Avet-Loiseau****et al., 2002** [21]	46 pPCL/147 MGUS/39 SMM/669 MM	FISH	Higher proportion of t(11;14), t(14;16), and 13q deletions in pPCL.
**Tiedemann et al., 2008** [8]	41 pPCL/39 sPCL/439 MM	FISH, conventional karyotyping, methylation-sensitive PCR, *TP53*, and *N/K-RAS* DNA sequencing	t(11;14) significantly more frequent in pPCL than in sPCL. High proportion of del(17p), *TP53* mutation, and biallelic inactivation in pPCL and sPCL.
**Chang et al.,** **2009** [22]	15 pPCL/26 sPCL/220 MM	cIg-FISH, FISH	del(13q), del (17p), t(4;14), 1q21 amplification and del(1p21) significantly more common in PCL than in MM. t(4;14) and del(1p21) associated with shorter OS. In multivariant analysis, t(4;14) remained a significant predictor for adverse OS in PCL.
**Chiecchio et al.,** **2009** [23]	10 pPCL/2 sPCL/861 MM	FISH, conventional karyotyping, aCGH, qRT-PCR	t(11;14) and t(14;16) significantly more frequent in PCL. Structural and numerical abnormalities frequently involve 8q24. *MYC* upregulation in PCL.
**Pagano****et al.,** **2011** [24]	73 pPCL (41 FISH), 53 sPCL	Conventional karyotyping (*n* = 28), FISH (*n* = 23)	Unfavorable cytogenetics: 56%.
**Usmani et al.,** **2012** [25]	13 pPCL/19 sPCL/1018 MM	GEP, FISH	GEP analyses distinguished pPCL from MM based on 203 gene probes.
**Lionetti et al.,** **2013** [26]	18 pPCL	FISH, GEP, SNP arrays, miRNA microarrays	83 deregulated miRNAs in pPCL compared to MM. Expression levels of miR-497, miR-106b, miR-181a, and miR-181b correlated with treatment response, and of miR-92a, miR-330-3p, miR-22, and miR-146a correlated with clinical outcome.
**Mosca et al.,** **2013** [27]	23 pPCL	FISH, SNP array, and GEP	Predominance of t(11;14) (40%) and t(14;16) (30%) Absence of activating mutations of *N/KRAS* in pPCL. GEP analysis revealed deregulated genes involved in metabolic processes.
**Todoerti et al.,** **2013** [28]	21 pPCL/55 MM	GEP	503-gene transcriptional signature distinguishes pPCL from MM. Underexpression of *YIPF6, EDEM3*, and *CYB5D2* associated with nonresponder pPCL. 27-gene model identifies pPCL patients with shorter OS.
**Cifola****et al.,** **2015** [29]	12 pPCL	WES	First study of mutational pattern in pPCL patients using WES. Identification of 14 candidate cancer driver genes, mainly involved in cell cycle, genome stability, RNA metabolism, and protein folding.
**Lionetti et al.,** **2015** [30]	24 pPCL/11 sPCL/132 MM	Targeted NGS for *BRAF* (exons 11 and 15), *NRAS* (exons 2 and 3) and *KRAS* (exons 2–4)	*MAPK* pathway affected in 42% pPCL, 64% sPCL, and 60% MM. *BRAF* mutations in 21% pPCL, 9% sPCL and 11% MM.
**Ronchetti****et al.,** **2016** [31]	24 pPCL/12 sPCL/170 MM/33 SMM/20 MGUS/9 NPC	lncRNA expression profiling by arrays	15 lncRNAs progressively increased, and six decreased from normal PCs to MGUS, SMM, MM, and PCL samples.
**Lionetti et al.,** **2016** [32]	12 pPCL/10 sPCL/129 MM	Targeted NGS for *TP53* (exons 4–9)	*TP53* mutations in 25% pPCL, 20% sPCL and 3% MM. del(17p) in 29% pPCL, 44% sPCL, and 5% MM. *TP53* mutations and del(17p) are markers of progression.
**Todoerti et al.,** **2018** [33]	14 pPCL/60 MM/5 MGUS	Global methylation patterns by high-density arrays	Global hypomethylation profile in pPCL. Decreasing methylation levels from MGUS to MM and pPCL.
**Rojas et al.,** **2019** [34]	9 pPCL/ 10 MM	Transcriptome arrays	Different transcriptome profiles between pPCL and MM carrying del(17p). RNA splicing machinery was one of the most deregulated processes in pPCL.
**Yu et al.,** **2020** [35]	46 pPCL	Conventional karyotyping (*n* = 34) and FISH (*n* = 37)	Predominance of del(13q) (38%), 1q gains (30%), del(17p) (27%), and t(11;14) (24%). t(4;14): not found.
**Schinke et al.,** **2020** [36]	23 pPCL/1273 MM	FISH, WES, and GEP	Predominance of complex structural changes and high-risk mutational patterns in pPCL. Driver genes with more mutations in pPCL than in MM: *KRAS*, *TP53*, *EGR1*, *LTB*, *PRDM1*, *EP300*, *NF1*, *PIK3CA*, and *ZFP36L1*.
**Nandakumar et al., 2021** [37]	68 pPCL (defined by ≥5% of clonal circulating PC)	FISH (*n* = 58)	Predominance of t(11;14) (47%), del(17p) (28%) and t(14;16) (12%).
**Todoerti et al.,** **2021** [38]	15 pPCL/50 MM	GEP, FISH	Different transcriptome profiles between pPCL and MM carrying t(11;14).
**Bútová et al.,** **2021** [39]	12 pPCL/11 sPCL/34 MM	lncRNA expression profile by NGS. Validation with qRT-PCR	13 deregulated lncRNAs between PCL and MM. Downregulation of LY86-AS1 and VIM-AS1 in PCL compared to MM.
**Papadhimitriou et al.,** **2022** [40]	25 pPCL/19 sPCL/965 MM	FISH and NGF	Distinct cytogenetic profile between pPCL and sPCL, predominantly more del(13q) (95%) and del(17p) (68%) in sPCL than in pPCL, but t(11;14) only detected in pPCL and MM cases, and significantly higher incidence of 8q24 rearrangements in pPCL (40%) compared to sPCL (26%) and MM (9%).
**Cazaubiel et al., 2020** [41] **a****nd** **Cazaubiel et al., 2022** [42]	96 pPCL/907 MM	Targeted NGS, RNA-seq, and FISH	*TP53* and *IRF4* mutations significantly more frequent in pPCL. Increased proportion of double hit profiles in pPCL. Different transcriptome profiles between pPCL with and without t(11;14).

FISH—fluorescence in situ hybridization; CGH—comparative genomic hybridization; aCGH—comparative genome hybridization arrays; NGS—next-generation sequencing; WES—whole-exome sequencing; SNP—single nucleotide polymorphism; GEP—gene expression profiling by microarrays; cIg-FISH—Cytoplasm light chain immunofluorescence with simultaneous interphase fluorescence in situ hybridization; qRT-PCR—quantitative real-time PCR; OS—overall survival; NGF—next-generation flow cytometry. * Only results related to genetic/genomic alterations are summarized.

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
