# Peer review of "Genomics of Plasma Cell Leukemia"

_cancers, 2022, doi:10.3390/cancers14061594_

Round 1

Reviewer 1 Report

Major comment: No references related to the possible PCL treatment modalities are represented in the manuscript.

Cell cycle genes co-expression can be added to the manuscript Cell cycle genes co-expression in multiple myeloma and plasma cell leukemia - PubMed (nih.gov)

Minor comment: Some excessive symbols are in Tab 1. Column “Summary of Results”

Author Response

We would like to thanks to the reviewer for the comments.

  • Major comment: No references related to the possible PCL treatment modalities are represented in the manuscript.

The present manuscript is only intended to review the genomic abnormalities of PCL. Indeed, the authors have expertise in analyzing PCL by genomic methodologies, but we are not currently involved in the clinical and therapeutic management of patients with PCL.

  • Cell cycle genes co-expression can be added to the manuscript Cell cycle genes co-expression in multiple myeloma and plasma cell leukemia - PubMed (nih.gov)

 We would like to thank the reviewer for providing us with this interesting information. Accordingly, we have added the following paragraph in page 7, line 226:

In this regard, a significant overexpression of CDKN2A, CCND3 and CCND1 genes, using quantitative RT-PCR, has been reported in PCL compared to MM samples, indicating a marked cell cycle dysregulation in the transition from MM to PCL [48].

  • Minor comment: Some excessive symbols are in Tab 1. Column “Summary of Results”

We have deleted the symbols.

Reviewer 2 Report

In manuscript cancers-1601331 Rojas and Gutierrez review current literature on genomics of plasma cell leukemia which indeed is of interest. The text is well structured, quite comprehensive and reads fluently. Thus, I recommend publication after minor revisions (see below).

The authors should try to incorporate very recent relevant studies (e.g. Cazaubiel et al. Blood 2022, Papadhimitriou et al. Biomedicines 2022) which were published during the review process.

All acronyms should be explained, e.g. CAR-T, MGUS, SMM, PFS etc.

Gene loci should be written in italics.

Since several typos (e.g. unnecessary “ü”s in Table 1) and errors occur, I recommend careful proof-reading of the final text.

Author Response

In manuscript cancers-1601331 Rojas and Gutierrez review current literature on genomics of plasma cell leukemia which indeed is of interest. The text is well structured, quite comprehensive and reads fluently. Thus, I recommend publication after minor revisions (see below).

The authors should try to incorporate very recent relevant studies (e.g. Cazaubiel et al. Blood 2022, Papadhimitriou et al. Biomedicines 2022) which were published during the review process.

All acronyms should be explained, e.g. CAR-T, MGUS, SMM, PFS etc.

Gene loci should be written in italics.

Since several typos (e.g. unnecessary “ü”s in Table 1) and errors occur, I recommend careful proof-reading of the final text.

Many thanks for all the comments. Accordingly, we have incorporated into the manuscript the two studies suggested by the reviewer (Cazaubiel et al. Blood 2022, and Papadhimitriou et al. Biomedicine 2022). Moreover, the typographic errors have been corrected and the symbols in table 1 have been deleted.